# Development of the ^99m^Tc-Labelled SST_2_ Antagonist TECANT-1 for a First-in-Man Multicentre Clinical Study

**DOI:** 10.3390/pharmaceutics15030885

**Published:** 2023-03-09

**Authors:** Doroteja Novak, Barbara Janota, Anton Amadeus Hörmann, Agnieszka Sawicka, Marko Kroselj, Alicja Hubalewska-Dydejczyk, Melpomeni Fani, Renata Mikolajczak, Petra Kolenc, Clemens Decristoforo, Piotr Garnuszek

**Affiliations:** 1The Department of Pharmaceutical Chemistry, Faculty of Pharmacy, University of Ljubljana, 1000 Ljubljana, Slovenia; 2Radioisotope Centre POLATOM, National Centre for Nuclear Research, 05-400 Otwock, Poland; 3Department of Nuclear Medicine, Medical University Innsbruck, 6020 Innsbruck, Austria; 4Department of Nuclear Medicine, University Medical Centre Ljubljana, 1000 Ljubljana, Slovenia; 5The Department of Pharmaceutical Technology, Faculty of Pharmacy, University of Ljubljana, 1000 Ljubljana, Slovenia; 6Chair and Department of Endocrinology, Jagiellonian University Medical College, 30-688 Cracow, Poland; 7Division of Radiopharmaceutical Chemistry, University Hospital Basel, 4031 Basel, Switzerland

**Keywords:** technetium-99m, somatostatin receptor antagonists, kit formulation, single dose toxicity, SPECT, radiopharmaceuticals, neuroendocrine neoplasms

## Abstract

Broad availability and cost-effectiveness of ^99^Mo/^99m^Tc generators worldwide support the use, and thus the development, of novel ^99m^Tc-labelled radiopharmaceuticals. In recent years, preclinical and clinical developments for neuroendocrine neoplasms patient management focused on somatostatin receptor subtype 2 (SST_2_) antagonists, mainly due to their superiority in SST_2_-tumour targeting and improved diagnostic sensitivity over agonists. The goal of this work was to provide a reliable method for facile preparation of a ^99m^Tc-labelled SST_2_ antagonist, [^99m^Tc]Tc-TECANT-1, in a hospital radiopharmacy setting, suitable for a multi-centre clinical trial. To ensure successful and reproducible on-site preparation of the radiopharmaceutical for human use shortly before administration, a freeze-dried three-vial kit was developed. The final composition of the kit was established based on the radiolabelling results obtained during the optimisation process, in which variables such as precursor content, pH and buffer, as well as kit formulations, were tested. Finally, the prepared GMP-grade batches met all predefined specification parameters together with long-term kit stability and stability of the product [^99m^Tc]Tc-TECANT-1. Furthermore, the selected precursor content complies with micro-dosing, based on an extended single-dose toxicity study, where histopathology NOEL was established at 0.5 mg/kg BW, being more than 1000 times higher than the planned human dose of 20 µg. In conclusion, [^99m^Tc]Tc-TECANT-1 is suitable to be advanced into a first-in-human clinical trial.

## 1. Introduction

Over the course of the last two decades, the imaging and treatment of patients with neuroendocrine neoplasms (NEN) have been redefined with the successful introduction of radiolabelled somatostatin analogues targeting overexpressed somatostatin receptors (SST), especially the somatostatin subtype 2 receptor (SST_2_) [1,2]. While the routine nuclear medicine practice currently relies on SST_2_-targeting radiopharmaceuticals that act as agonists, namely [^99m^Tc]Tc-EDDA/HYNIC-TOC, [^177^Lu]Lu-DOTATATE or [^68^Ga]Ga-DOTATOC, the preclinical and clinical development is focused on SST_2_ antagonists [3,4,5,6,7]. This is due to the ability of antagonists to recognise more binding sites on the receptor [4,8]. This resulted in higher tumour uptake, superior tumour-to-background contrast and thus enhanced image sensitivity, and higher tumour radiation doses [6,7]. The first phase I/II and pilot clinical studies on ^68^Ga-labelled SST_2_-antagonist LM3 (p-Cl-Phe-cyclo(D-Cys-Tyr-DAph(Cbm)-Lys-Thr-Cys)-D-Tyr-NH_2_) confirmed its high potential for PET/CT imaging [9,10]. Even though the role of ^99m^Tc- or ^68^Ga-labelled SST-agonists, especially with the emergence of ^68^Ga-labelling kits (SomaKit TOC^®^ and NETSPOT™), remains undisputed as an established diagnostic tool for molecular imaging of NENs [11,12], ^99m^Tc-labelled SST_2_-antagonist for SPECT/CT may beneficially impact in the current standard of care of NEN; expanding diagnostic imaging capabilities for disease staging and treatment efficacy monitoring and therefore making it available to a larger patient population. Several factors, including the much wider availability of ^99^Mo/^99m^Tc generators worldwide, cost-effectiveness, favourable physical properties, especially the half-life of 6 h and optimal γ-energy of 140 keV [13], make ^99m^Tc the radionuclide of choice for many institutions and support the development of novel ^99m^Tc-labelled radiopharmaceuticals. These facts, combined with the improved image sensitivity of SST_2_-antagonists over agonists, argue for the use of a ^99m^Tc-labelled SST_2_-antagonist to further advance personalised medicine in the diagnosis and management of NENs. Preclinical developments showed that the SST_2_-antagonists conjugated to the tetraamine-chelator 6-carboxy-1,4,8,11-tetraazaundecane (N4) exhibited high SST_2_ uptake in vitro and in vivo, while when conjugated to HYNIC (6-hydrazinopyridine-3-carboxylic acid) the introduction of a spacer is unavoidable and, even in this case modifications are needed for reaching a modest SST_2_-mediated uptake [14,15]. Therefore, the N4-chelator was chosen for conjugation to two promising SST_2_-antagonists: LM3 and p-Cl-BASS (p-Cl-Phecyclo(D-Cys-Tyr-D-Trp-Lys-Thr-Cys)-D-Tyr-NH_2_) at the N-terminal for ^99m^Tc-labelling [16]. Based on the head-to-head comparison of ^99m^Tc-labelled N4-LM3 (TECANT-1) and N4-p-Cl-BASS (TECANT-2) as described in our previous work [16], TECANT-1 was selected as the optimal candidate for conducting the first clinical trial with a ^99m^Tc-labelled SST_2_-antagonist for imaging of NENs. In order to ensure a high quality of on-site preparation and a harmonised production in the clinical centres involved, a freeze-dried kit formulation for radiolabelling of the precursor N4-LM3 with ^99m^Tc was developed. Herein we report on the chemical and pharmaceutical development, characterisation of the kit formulation, as well as the toxicity study to support a multi-centre clinical trial.

## 2. Materials and Methods

### 2.1. Chemicals and Materials

**Wet radiolabelling**. If not otherwise specified, chemicals, materials, and solvents were of reagent quality and were purchased from Merck (Darmstadt, Germany), Sigma Aldrich (Steinheim, Germany) or Honeywell (North Carolina, United States) and used without additional purification. Water for injection (Ph. Eur.) was provided by Fresenius Kabi (Halden, Norway).

**Kit preparation**. If not otherwise stated, chemicals, materials, and solvents were of pharmaceutical grade and were used without further purification. Water for injection (Ph. Eur.) was provided by Bieffe Medital (Grosotto, Italy). For lyophilisation, 2 mL vials made from colourless, high hydrolytic resistance class, neutral borosilicate glass FIOLAX^®^ (Ph. Eur. Type I) (SCHOTT, Müllheim, Germany), closed with chlorobutyl rubber closures (West Pharmaceutical Services Deutschland, Eschweiler, Germany) and sealed with aluminium caps (West Pharmaceutical Services Deutschland, Eschweiler, Germany) were used.

**TECANT-1 and TECANT-2 precursors.** TECANT-1 (N4-p-Cl-Phe-cyclo(D-Cys-Tyr-D-Aph(Cbm)-Lys-Thr-Cys)-D-Tyr-NH_2_, where D-Aph(Cbm): D-4-amino-carbamoyl-phenylalanine) and TECANT-2 (N4-p-Cl-Phe-cyclo(D-Cys-Tyr-D-Trp-Lys-Thr-Cys)-D-Tyr-NH_2_) were custom made by piChem (Raaba-Grambach, Austria). TECANT-1 was produced in GMP-grade quality. The chemical structure of [^99m^Tc]Tc-TECANT-1 is shown in Figure 1.

**[^99m^Tc]TcO_4_^−^ eluate.** For radiolabelling, a solution of Na^+^[^99m^Tc]TcO_4_^−^ was obtained from a commercial ^99^Mo/^99m^Tc-generator (Polatom, Otwock, Poland, or Curium, Petten, The Netherlands). The properties of the [^99m^Tc]TcO_4_^−^ eluate are in accordance with the European Pharmacopoeia current edition, monograph 0124. The radiochemical purity of [^99m^Tc]TcO_4_^−^ eluate was >98% with radionuclidic impurity not more than: 0.1% ^99^Mo, 5 × 10^−3^% ^131^I, 5 × 10^−3^% ^103^Ru, 6 × 10^−5^% ^89^Sr, 6 × 10^−6^% ^90^Sr. Clear and colourless [^99m^Tc]TcO_4_^−^ eluate was sterile, and the pH was in the range of 5.5–7.5.

### 2.2. Analytical Methods

**Radio-reversed phase (RP)HPLC**. The radio-(RP)HPLC method reported in Appendix A was used for both identity and the determination of the percentage of [^99m^Tc]Tc-TECANT-1 (% [^99m^Tc]Tc-TECANT-1 (HPLC)) defined as the area of the peak corresponding to [^99m^Tc]Tc-TECANT-1 relative to the area of all radioactive peaks in the radio-(RP)HPLC chromatogram.

**iTLC.** For the determination of the content of ^99m^Tc-colloid species after radiolabelling, the iTLC system described in Appendix A was used.

**Calculation of the radiochemical purity (RCP).** The radiochemical purity was calculated using the following formula:$$\;\mathrm{RCP}=\frac{\left(100-B\right)\times T}{100}$$

B = percentage of radioactivity corresponding to ^99m^Tc-colloid species determined in the iTLC test as described under iTLC

T = area of the peak corresponding to [^99m^Tc]Tc-TECANT-1 in the radio-(RP)HPLC chromatogram (% [^99m^Tc]Tc-TECANT-1 (HPLC)).

Validation of the HPLC and iTLC methods for the determination of % [^99m^Tc]Tc-TECANT-1, as well as RCP, included the assessment of accuracy, precision, specificity, limit of quantification, linearity, range and robustness (see also Table 1 and Table 2).

The [^99m^Tc]TcO_4_^−^ content after radiolabelling was determined during the assessment of development batches using the iTLC technique detailed in Appendix A.

**Inductively coupled plasma—optical emission spectrometry ICP-OES.** The determination of total tin in the TECANT-1 kit for ^99m^Tc-labelling was carried out using optical emission spectrometry with inductively coupled plasma (ICP-OES Optima 7300DV spectrometer). The method has been validated and approved for use in routine analysis of TECANT-1.

**Bacterial endotoxins**. Endotoxins testing was performed by gel-clot method according to Ph. Eur. (monograph 20614) [17] using an endotoxin standard control (KSE): *E. Coli* 055:B5 (Charles River, South Carolina, United States).

**Sterility testing.** Sterility testing was performed according to the Ph. Eur. (monograph 20601) using the direct inoculation method [18].

**Water content.** For the determination of residual water in kit formulations, the coulometric Karl-Fisher-oven method was used as described in Ph. Eur. (monograph 20532) [19].

### 2.3. Study Design

To ensure the traceability of the final clinical product using the same GMP-grade batch of TECANT-1 as in the development stage, the earlier work on robust optimisation of the composition of the kit as well as first kit-labelling (batch 05a/20) were carried out with the use of TECANT-2, a variation of TECANT-1 with substitution in position 4 D-Trp instead of D-Aph(Cbm), but identical labelling behaviour [16]. After robust optimisation and confirmation of the feasibility of the kit development, all subsequent batches used for the fine optimisation of the kit formulation, determination of specifications, and stability studies contained TECANT-1.

### 2.4. Preformulation Studies

**Radiolabelling optimisation.** Wet radiolabelling of TECANT-1/TECANT-2 was performed in analogy to radiolabelling of Demotate with [^99m^Tc]TcO_4_^−^ published previously [20]. Shortly, a freshly prepared stock solution of TECANT-1/TECANT-2 peptide was prepared by dissolving the peptide in water for injection (1 µg/µL), aliquoted prior to the radiolabelling and stored at −80 °C until further use. Radiolabelling was performed in a pre-lubricated low binding Eppendorf^®^ (Hamburg, Germany) vial, to which 25 µL of phosphate buffer (0.5 M, pH 11) was added, followed by the addition of trisodium citrate dihydrate (0.1 M) (amounts varying from 0.13 to 0.39 mg) and 500 µL of [^99m^Tc]TcO_4_^−^ (400–600 MBq). To this solution, freshly thawed precursor stock solution (amounts varying from 10–50 µg) and SnCl_2_ × 2H_2_O (amounts varying from 5–20 µg from a 2 mg/mL stock solution in 99% EtOH) were added and purged with nitrogen prior to use. Subsequently, the radiolabelling mixture was incubated at ambient temperature for varying times (10 to 30 min). Afterwards, the radiolabelling solution was neutralised by the addition of NaH_2_PO_4_ (15 µL, 1 M). Quality control of the radiolabelled product and RCP determination was performed by radio-(RP)HPLC and iTLC, as described above.

### 2.5. Kit Preparation

Development batches (batch no. 2–10 of TECANT-1/TECANT-2 kits) were prepared with varying different parameters, including the number of kit vials (n = 2 and 3), precursor content (10–50 µg) and SnCl_2_ × 2 H_2_O content (15 and 20 µg). Then, three batches (batch no. 01B/21, 02B/21 and TEC-01/01/22) with optimised compositions were prepared. The manufacturing of the TECANT-1 kit for the investigational medicinal product was carried out in clean rooms in compliance with cGMP requirements.

The bulk solution of the formulation for each kit vial was prepared in a clean room using pre-sterilised glass vials, containers and stoppers.

Kit preparation is exemplified below for the 3-vial kit; other formulations were prepared accordingly.

**Vial 1.** The bulk manufacturing started with the transfer of approximately, but not more than, 90 mL of water for injection into a plastic bottle with a stir bar in place. TECANT-1 (2 mg net peptide) was added to the water while stirring and purging with nitrogen gas. The concentration of TECANT-1 was determined using HPLC. Next, 13 mg of trisodium citrate dihydrate was added to the solution. Then 1.5 mg of SnCl_2_ × 2 H_2_O, dissolved in 1.5 mL EtOH, purged with nitrogen gas, was added to the bulk solution under continuous stirring. Finally, the bulk solution was diluted with water for injection to the final volume of 100 mL, under continuous stirring and purged with nitrogen gas.

**Vial 2.** The bulk manufacturing of kit vial 2 was prepared in a similar manner as vial 1, beginning with the transfer of approximately, but not more than, 90 mL of water for injection into a plastic bottle with a stir bar in place. Next, 450 mg of Na_2_HPO_4_ × 12H_2_O and 40 mg of NaOH were then added to the water under stirring. The pH of the bulk solution was adjusted to 11.4–11.6 with 1 M hydrochloric acid or 1 M sodium hydroxide, if necessary. Finally, the bulk solution was diluted with water for injection to the final volume of 100 mL, under continuous stirring.

**Vial 3.** The bulk manufacturing of the kit vial 3 was prepared in a similar manner as vial 1, starting with the transfer of approximately, but not more than, 90 mL of water for injection into a plastic bottle with a stir bar in place. Next, 156 mg of NaH_2_PO_4_ × 2H_2_O were added to the water while stirring. The pH of the bulk solution was adjusted to a pH of 4.5–5.5 with 1 M hydrochloric acid or 1 M sodium hydroxide, if necessary. Finally, the bulk solution was diluted with water for injection to the final volume of 100 mL, under continuous stirring.

The bulk solutions were then sterilised by separate filtration through a sterile PALL Mini Kleenpak™ Syringe filter (0.2 µm Supor membrane) (Pall Cooperation, New York, USA) and dispensed into the 2 mL vials within the range of 1 mL ± 1% per vial and covered with a rubber stopper. The vials were then transferred into the freeze-dryer. Afterwards, the product was lyophilised according to the lyophilisation scheme presented in Appendix A.

The vials were sealed under sterile conditions in a nitrogen atmosphere (with a slight overpressure) with rubber stoppers, taken out of the freeze-dryer and capped with metal caps. The vials were thoroughly inspected visually, and quality control followed, including identification, precursor quantification by (RP)HPLC analysis (UV-Vis at 220 nm), SnCl_2_ × 2H_2_O content determination by ICP-OES, pH measurement after reconstitution with 1 mL of water for injection, water content determination by the Karl-Fischer- method, sterility testing and determination of endotoxins content.

### 2.6. Kit Radiolabelling

**Two-vial kit.** Vial 1 was reconstituted with 500 µL of [^99m^Tc]TcO_4_^−^ eluate (500–600 MBq), and the reaction mixture was incubated at room temperature for 30 min. Meanwhile, vial 2 was reconstituted with 500 µL water for injection. After the incubation was completed, the content of vial 2 was transferred to vial 1 to neutralise the reaction mixture. Quality control of the radiolabelled product—determination of RCP—was performed by radio-(RP)HPLC and iTLC as described above.

**Three-vial kit**. Vial 2 was reconstituted with 500 µL water for injection, and the entire content was then transferred to vial 1. To this solution, 500 µL of [^99m^Tc]TcO_4_^−^ eluate (500–600 MBq) was added, and the reaction mixture was incubated at room temperature for 30 min. Meanwhile, vial 3 was reconstituted with 500 µL of water for injection. After the incubation was completed, the content of vial 3 was transferred to vial 1 to neutralise the reaction mixture. Quality control of the radiolabelled product—determination of RCP—was performed by radio-(RP)HPLC and iTLC as described above.

### 2.7. Stability Studies

**Long-term stability study.** The stability of the freeze-dried kits stored at 5 ± 3 °C in the primary container tightly closed with a rubber stopper and an aluminium seal was assessed for a period of 12 months. The kits were visually inspected at different time intervals (at release and at 3, 6, 9 and 12 months after release), while the identity, assay of TECANT-1 and SnCl_2_, pH of each vial after reconstitution with 1 mL of water for injection, determination of residual water in the lyophilisate, sterility and bacterial endotoxin content were assayed at the endpoint. The ^99m^Tc-labelling was carried out at each time point following the established protocol. Determination of RCP was performed by radio-RP-HPLC and iTLC as described above.

**Stability study of the radiolabelled compound after kit radiolabelling.** The stability of the final radiopharmaceutical preparation was assessed based on RCP and evaluated using the radio-(RP)HPLC and iTLC described above. After radiolabelling, the before-mentioned parameters were evaluated directly after preparation, as well as after 1, 2 and 4 h of storage in an upright posture at room temperature. To test the robustness of the radiolabelled preparation, different amounts of radioactivity and volume were evaluated with regard to the stability of the radiolabelled [^99m^Tc]Tc-TECANT-1 preparation.

### 2.8. Toxicity of TECANT- 1

**Animals.** Female CD-1 mice of 8–9 weeks of age and a body weight of 23.6–35.4 g were used for extended single-dose toxicity studies. Procedures and facilities involving handling testing animals comply with the requirements of Directive 2010/63/EU and the German animal welfare and drug legislation.

**Extended single-dose toxicity.** An extended single-dose toxicity study, including a 14-day recovery period, was conducted by BSL Bioservice (Munich, Germany) in accordance with internal SOPs and internationally accepted guidelines and recommendations, including ICH Guideline M3(R2) on non-clinical safety studies for the conduction of human clinical trials and marketing authorisation for pharmaceuticals (EMA/CPMP/ICH/286/1995) [21] CPMP/SWP/1042/99 Rev 1, Guideline on Repeated Dose Toxicity [22], and ICH Guideline CPMP/ICH/539/00–ICH S7A, Safety Pharmacology Studies for Human Pharmaceuticals, Adopted by CPMP [23].

In short, TECANT-1 precursor was administered by a single slow intravenous (bolus) injection to CD-1 mice, followed by a 24-h observation period (main study). In addition, the reversibility or progression of treatment-related changes or any delayed toxicity was assessed 13 days after the test item administration (Recovery Study). In total, 45 CD-1 mice were distributed into three dose groups, each containing 15 animals. The animals from Group C received only phosphate-buffered saline (PBS), serving as controls, while Groups LD and HD received the TECANT-1 precursor solutions (prepared in PBS, application volume: 5 mL/kg body weight) by single intravenous (bolus) injection at a low (0.1 mg/kg body weight) and a high dose (0.5 mg/kg body weight), respectively. Ten animals per group were euthanised one day after administration of the test item (Main Study), while the remaining 5 animals per group were sacrificed 13 days after test item administration (Recovery Study). At the end of the observation period (24 h or 13 days), samples were collected to evaluate haematology, clinical chemistry, necropsy and histopathology data. The animals in the control group were handled in an identical manner to the test group subjects.

## 3. Results

### 3.1. Analytical Methods Development

Characterisation of impurities was achieved by radio-(RP)HPLC and iTLC with the aid of suitable radioactive control samples: [^99m^Tc]TcO_4_^−^ eluate (“free” ^99m^Tc), [^99m^Tc]Tc-citrate, and artificially produced ^99m^Tc-colloid species.

**Radio-(RP)HPLC.** The radiochemical identity of [^99m^Tc]Tc-TECANT-1 was determined by comparison to the external TECANT-1 reference standard. The respective retention times (R_t_) of radioactive control impurities were:

[^99m^Tc]TcO_4_^−^: R_t_ = 0.97 min

[^99m^Tc]Tc-citrate: R_t_ = 0.96 min

^99m^Tc-colloid species: not visible

Impurities related to radiolytic side products of [^99m^Tc]Tc-TECANT-1 could be detectable at a retention time of ≥6 min.

Representative chromatograms are shown in Figure 2.

Validation of the radio-(RP)HPLC system was performed according to the EANM guideline for analytical validation of radiopharmaceuticals [24], and the specifications, acceptance criteria and results are reported in Table 1.

**iTLC.** The validation parameters, acceptance criteria and results for the iTLC method using 5 M ammonium acetate buffer:methanol (1:1) (*v*/*v*) as mobile phase are reported in Table 2; a representative chromatogram is shown in Appendix A.

### 3.2. Preformulation Studies

**Optimisation of excipients content.** To initially assess the optimal amount of SnCl_2_ × 2H_2_O and trisodium citrate dihydrate, preliminary radiolabelling studies using TECANT-2 were carried out. The radiolabelling was performed at room temperature for 30 min at a fixed amount of TECANT-2 (10 µg) using 400 MBq [^99m^Tc]TcO_4_^−^. Table 3 summarises the influence of the different amounts of SnCl_2_ × 2H_2_O and trisodium citrate dihydrate on radiochemical purity. Based on the results, 15 µg of SnCl_2_ × 2H_2_O and 0.13 mg trisodium citrate dihydrate were selected as suitable starting amounts for further optimisation aiming at high RCP.

**Optimisation of kit radiolabelling conditions.** In order to further optimise the wet-radiolabelling conditions of TECANT-1, different parameters, such as the activity of [^99m^Tc]TcO_4_^−^ added, pH and reaction time, were tested (Appendix A). Radiolabelling could be performed at high apparent molar activities and overall high RCP. When different amounts of TECANT-1 (10, 15, and 50 µg) were tested, a decrease in RCP was observed for 10 and 15 µg over a 30 min incubation period (93.9 ± 4.8% and 95.5 ± 1.2% for 10 and 15 µg, respectively), while the highest RCP was achieved with shorter incubation times, resulting in more than 97% RCP after 10–15 min for both amounts of peptide. For radiolabelling solutions containing 50 µg TECANT-1, a higher RCP was detected at later time points (93.8 ± 2.0% at 10 min vs. 95.8 ± 2.2% at 30 min). Increasing the amount of radioactivity added to the radiolabelling solution resulted in a slight decrease of RCP. Promising purities (95.1 ± 1.6%) could be achieved using about 700 MBq or less of [^99m^Tc]TcO_4_^−^ at 30-min incubation time for all preparations, regardless of the quantity of the precursor. When using more than 700 MBq (700–1200 MBq), the RCP dropped to 91.1 ± 2.8% (30 min incubation). Therefore, the optimal incubation time seemed to be in the range of 15–20 min. When the pH was lowered to 8 during radiolabelling, the labelling kinetics slowed down (76.3%, 10 min) but increased to almost quantitative yields after 30 min (98.0%). Figure 3 shows the % [^99m^Tc]Tc-TECANT-1 of different amounts of TECANT-1 when radiolabelled with [^99m^Tc]TcO_4_^−^ (<700 MBq) over a period of 30 min as well as the influence of labelling pH on labelling kinetics.

Based on the results obtained in the optimisation of kit radiolabelling conditions, a simple and convenient two-vial kit formulation for ^99m^Tc-labelling was developed first.

### 3.3. Kit Preparation

**Two-vial kit.** As indicated under “Section 3.2. Preformulation studies”, a two-vial kit formulation was first developed and tested under standard radiolabelling conditions (RT, 30 min). The first vial contained all components necessary for ^99m^Tc labelling, while the second vial contained a pH neutraliser. The RCP of the product using this type of kit did not exceed 90% (Appendix A). As a result, the two-vial kit was considered inappropriate.

**Three-vial kit.** A three-vial kit formulation was developed, separating the buffer components from the precursor, SnCl_2_ × 2H_2_O and trisodium citrate dihydrate. Initial radiolabelling results, as described in Table 4 (RT, 30 min incubation time), indicated that the three-vial kit increased RCP compared to the two-vial kit reaching RCP up to 94%.

The development batches 05–07/20 (TECANT-1 content varying among 10, 15 and 50 µg) were further labelled with ~600 MBq of [^99m^Tc]TcO_4_^−^ at room temperature and at different incubation times (10–30 min) (all tested parameters are summarised in Appendix A). Almost quantitative yields (94.5 ± 2.0% to 96.4 ± 1.1%) were achieved for every batch (Appendix A), which is in line with the previous results after 30 min of incubation time.

The above results indicated that this type of kit seemed most appropriate for the intended application; therefore, a kit was produced under GMP conditions which contained 20 µg of TECANT-1 and all other components as in the test kits.

**Final kit composition.** The optimal pH value seemed to be 11; nevertheless, the labelling of TECANT-1 with [^99m^Tc]TcO_4_^−^ was still effective with a larger fluctuation of the pH to more acidic values, indicating a high robustness of the labelling reaction. For the radiolabelling, the incubation time seemed to be optimal in the range of 15–20 min, especially when higher activities were used. The final kit composition was established based on these radiolabelling results and is reported in Table 5.

**Specifications.** Three batches of kits (01B/21, 02B/21 and TEC-01/01/22) were produced using the final kit formulation (Table 6). Direct comparison of labelling batch 01B/21 using two different ^99^Mo/^99m^Tc-generators are shown in Appendix A with no significant differences between the two generator types.

### 3.4. Stability Studies

The batches 01B/21 and 02B/21 were tested for stability to determine the kit and radiopharmaceutical shelf-life.

**Long-term kit stability.** Samples of both batches 01B/21 and 02B/21 showed no changes in appearance within the observation period at any given time point (t = 0, 3, 6, 9, and 12 months). Additionally, no changes in pH after reconstitution with 1 mL water, water, TECANT-1 and tin content were detected at the latest time-point (12 months). All samples were radiolabelled after different storage times, and the results are summarised in Table 7.

**Stability of radiolabelled product.** Stability was tested for up to 4 h for both batches, and the results are shown in Table 8. To test the robustness of the radiolabelled kit solution, different amounts of activity and volume were evaluated with regard to the stability of the [^99m^Tc]Tc-TECANT-1 preparation. The RCP values did not decrease below the set limit of up to 4 h after radiolabelling.

### 3.5. Toxicity of TECANT-1

All animals survived until the end of the pre-scheduled study periods. Some observed changes were recognised as incidental, while none were considered test item related. Overall, there were no organ weight changes, macroscopic changes or histological alterations that could be attributed to TECANT-1 treatment.

There were no clinical signs observed, toxicologically relevant effects on body weight development, parameters of haematology or clinical biochemistry and urinalysis during the main or recovery phase of this study. The histopathological examination revealed no test item related histological alterations. Therefore, and under the conditions of this study, the histopathology NOEL (No Observed Effect Level) was established at 0.5 mg/kg BW.

## 4. Discussion

^99m^Tc-labelled SST_2_-antagonists is a valuable choice to ensure the wide availability of radiopharmaceuticals for molecular imaging of NENs—one of the critical steps in improving current patient care. The latter is the motivation behind the first-in-human multi-centre clinical trial proposed within the European research project TECANT, co-funded by the ERA-NET scheme of the European Union (Horizon 2020). However, advancing the potential candidate from the preclinical setting into a first-in-human use requires demanding and laborious work to ensure that all expected quality and safety standards for human investigation are met. As on-site preparation of radiopharmaceuticals is preferred and mandatory for the short-lived radionuclide ^99m^Tc to ensure the high quality of the administered product and to avoid any stability problems during, e.g., shipment, successful performance of the radiolabelling of the precursor is of utmost importance. A well-formulated, ready-to-use freeze-dried kit offers a convenient solution to ensure a rapid, robust and harmonised, yet simple, radiolabelling procedure in all clinical centres.

The preformulation studies were first carried out in order to narrow down the variations for the initial kit formulations starting from the literature radiolabelling procedures used for the N4-conjugated SST-agonist [^99m^Tc]Tc-N_4_-Tyr^3^-Octreotate [20]. Based on TECANT-2 wet radiolabelling, the optimal amounts of SnCl_2_ and trisodium citrate dihydrate were determined. The starting amount of precursor was set at 10 µg, which was based on the obtained purity determined by HPLC (above 88% for any given condition), and seemed to be sufficient to achieve high RCPs with simultaneous high apparent molar activities. At a fixed amount of trisodium citrate dihydrate (0.13 mg), the content of SnCl_2_ up to 15 µg ensured RCP > 93%, while a minor drop in RCP (88%) was observed with 20 µg. A slight decrease in RCP when gradually increasing the amount of trisodium citrate dihydrate could be observed, and therefore the amount of 0.13 mg was selected. No variation in other labelling conditions (reaction time, temperature or pH) was done in this initial optimisation phase step.

The two-vial kit composition was formulated based on these results, with precursor and all other excipients being part of one vial to allow only for the addition of [^99m^Tc]TcO_4_^−^ eluate, while the content of the second vial would serve only for the reconstitution of the neutraliser that would be added to the reaction mixture at the end of the reaction. Nonetheless, high values (>90%) of RCP could not be achieved using such a kit, and the composition was changed to the three-vial kit as the purity by HPLC analysis of initial radiolabelling ranging from 91.6% up to 99.0% pointed to the successful formulation of kit composition and labelling procedure. The crucial change in kit composition seemed to be the separation of labelling buffer, specifically the adjustment of pH of the eluate before the addition to the vial containing the active substance, SnCl_2_ and trisodium citrate dihydrate. The latter could be explained by an instability of TECANT-1 in the phosphate buffer matrix as tests on the chemical purity of TECANT-1 in phosphate buffer showed deterioration to 55% at 24 h (determined by HPLC, λ = 220 nm). The proposed three-vial kit formulation contains TECANT-1 in the amount of 20 μg (15 nmol) and additionally buffer substances as well as reductant SnCl_2_ and a small amount of ethanol. Although an amount of 10 µg TECANT-1 seemed to be sufficient to meet the required quality, the final amount of 20 µg of TECANT-1 was selected to ensure a reliable and robust radiolabelling procedure during the pharmaceutical translation. Additionally, lower amounts of peptides are very challenging in terms of pharmaceutics and handling with respect to adsorption and analytics, while a higher quantity does not seem to benefit the overall RCP. The large molar excess of precursor TECANT-1 compared to ^99m^Tc also provides additional safety by assisting in the complexing of practically all ^99m^Tc ions. These measures are standard strategies applied for routinely established ^99m^Tc-radiopharmaceutical kits. Buffer substances were used to provide suitable pH for radiolabelling of peptides and stability of the radiolabelled product, while the SnCl_2_ is necessary for the ^99m^Tc-labelling to take place. Additionally, all substances can be freeze-dried, allowing a formulation of a radiopharmaceutical kit with a long shelf life. All the additives have been used in a number of pharmaceutical formulations and therefore pose a very low risk when used in this application. The amount of ethanol used in the kit dispensing process was not specified, as the low amount of 15 µL per kit was removed in the freeze-drying process; this risk-based approach was accepted by the reviewing competent authorities. Technetium-99m in the form of [^99m^Tc]TcO_4_^−^ for radiolabelling was eluted from commercially available ^99^Mo/^99m^Tc generators with a marketing authorisation in EU countries; no significant differences were observed between the two different generator types.

The final composition of the kit was established based on the radiolabelling results and was successfully adapted for three GMP-grade batches, meeting all predefined specification parameters (based on several Ph. Eur. monographs and guidelines [24,25,26,27,28,29]), including those for the kit itself as well as those related to the radiolabelling. Based on the current results, the proposed shelf life of the TECANT-1 kit preparation is 12 months when stored at +4 °C (2 to 8 °C). The current formulation seems to also ensure the high stability of the radiopharmaceutical in solution up to 4 h after radiolabelling. Therefore, the shelf-life of the [^99m^Tc]Tc-TECANT-1 preparation was set to be 2 h after radiolabelling when stored at temperatures up to 25 °C, which is sufficient for this type of study while still ensuring a safety margin for different activity ranges. However, further development and studies may be needed towards the establishment of clinical routine, e.g., widening the activity range, adding stabilisers for potential longer stability and evaluating higher volumes used for radiolabelling.

The toxicology profile of the TECANT-1 was assessed according to ICH guideline M3(R2) as a single intravenous injection in one animal species [21]. The outbred CD-1 mice were selected as the rodent species for the purpose of this study as they are robust, larger, and better breeders than inbred mice and are preferable in assessing the responses throughout a population of heterogeneous individuals. Under the conditions of the study, the administered dose at a concentration over 1000 times higher than the planned human dose could be determined as NOEL. As [^99m^Tc]Tc-TECANT-1 repeated administration for patient monitoring is unlikely to occur, and the doses used in clinical practice are expected to be low and separated in time, following the micro-dose approach (total dose not exceeding 100 μg), no repeat dose toxicity studies were considered necessary. Furthermore, considering the favourable safety profile and planned administration regime (single low dose), no studies on fertility, embryology, mutagenicity or long-term carcinogenicity have been conducted or are planned to be conducted.

## 5. Conclusions

Within the TECANT project, a preselected ^99m^Tc-labelled SST_2_-antagonist TECANT-1 (also known as N4-LM3) was successfully developed into a kit formulation. The developed kit allows robust and reproducible radiolabelling with ^99m^Tc with consistently high overall RCP (>90%) in short time periods (10 min at RT) without the need for additional purification. The primary goal to provide a reliable and harmonised method for facile preparation of [^99m^Tc]Tc-TECANT-1 in quality that meets the release criteria for clinical studies and is amenable for use in a hospital radiopharmacy setting was therefore achieved. Together with the favourable toxicological and promising pharmacological profile, [^99m^Tc]Tc-TECANT-1 is readily available to be advanced into a first-in-human multicentre clinical trial.

## Figures and Tables

**Figure 1 pharmaceutics-15-00885-f001:**
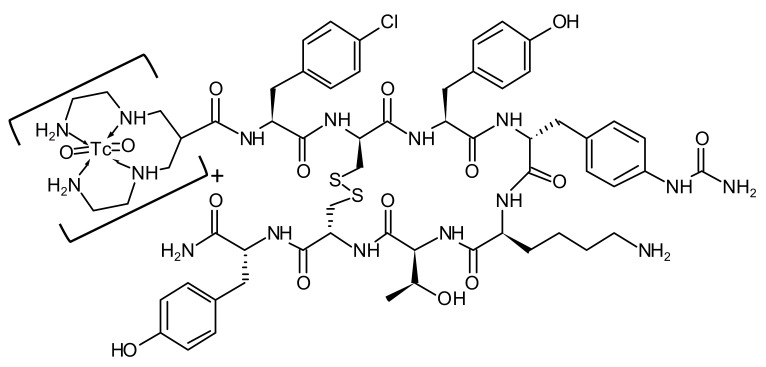
The chemical structure of [^99m^Tc]Tc-TECANT-1.

**Figure 2 pharmaceutics-15-00885-f002:**
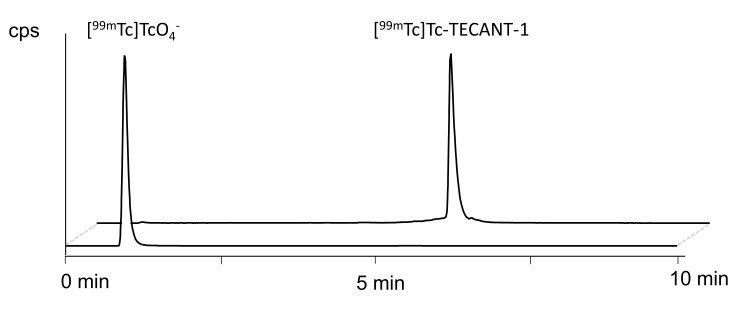
Representative radio-(RP)HPLC chromatogram of [^99m^Tc]Tc-TECANT-1 and [^99m^Tc]TcO_4_^−^ eluate.

**Figure 3 pharmaceutics-15-00885-f003:**
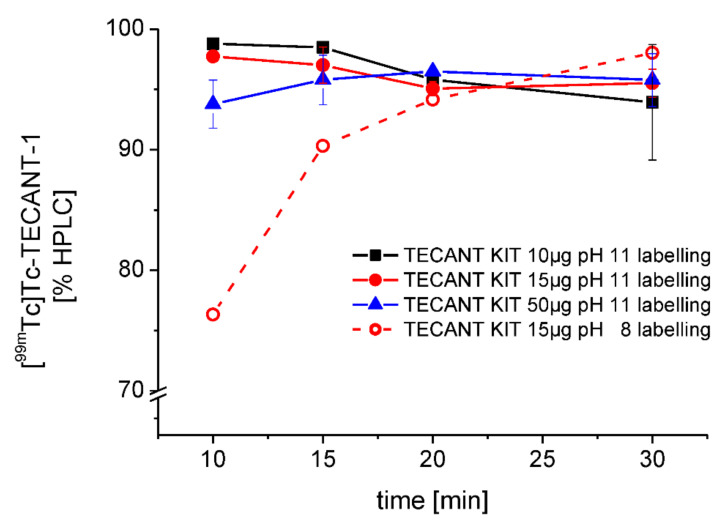
% of [^99m^Tc]Tc-TECANT-1 purity in dependence on the amount of the precursor, reaction time and the pH of the reaction solution, determined by HPLC.

**Table 1 pharmaceutics-15-00885-t001:** Summary table of validation parameters, acceptance criteria and results for radio-(RP)HPLC method for determination of identity and percentage of ^99m^Tc-labelled TECANT-1 (% [^99m^Tc]Tc-TECANT-1 (HPLC) (T)).

Parameter	Method	Acceptance Criteria	Result
Accuracy (Identity)	Comparison of R_t_ of TECANT-1 with R_t_ of [^99m^Tc]Tc-TECANT-1	R_t_ shift [^99m^Tc]Tc-TECANT-1 over TECANT-1 +10–20% (n ≥ 5)	R_t_ TECANT-1: 4.89 ± 0.04 min R_t_ [^99m^Tc]Tc-TECANT-1: 5.74 ± 0.07 min R_t_ shift: +17.6% (mean n = 6)
Specificity (Identity)	Resolution of TECANT-1 and TECANT-2	Resolution > 2 (n = 3)	R_t_ TECANT-1: 4.9 min R_t_ TECANT-2: 5.4 min Resolution (range): 2.94–3.45
Accuracy (T)	Recovery of [^99m^Tc]TcO_4_^−^ eluate of a sample (n = 6)	Recovery > 95%	98.67 ± 0.30%
Precision (T)	Sequential analysis of a sample of [^99m^Tc]TcO_4_^−^ (n = 6)	Area of peak CV ≤1% (n ≥ 6)	mean area (n = 8): 72.84 + 0.42 CV = 0.57%
Specificity (T)	Sequential analysis of a sample of [^99m^Tc]TcO_4_^−^ (n = 6)	R_t_ impurity CV < 5%	R_t_ impurity: 1.0 ± 0.006 min CV = 0.62%
LOQ (T)	Analysis of a dilution series of radioactivity, LOQ according to Ph. Eur. 10× height of background	LOQ < 0.5% of a peak with an activity from a sample with diluted 1:5 with 750 MBq/mL *	LOD = 0.45 kBq LOQ = 1.5 kBq ** LOQ 0.1% of a sample with minimum activity
Linearity (T)	Analysis of a dilution series of [^99m^Tc]TcO_4_^−^ (450 MBq/mL to 45 kBq/mL)	R^2^ > 0.99	0.9991
Range (T)	Analysis of a dilution series (450 MBq/mL to 45 kBq/mL)	Linearity within 1:10,000 dilution	1:10,000
Robustness	R_t_ variation on different day and with different operator	CV ≤ 1% (n ≥ 6)	Original R_t_ [^99m^Tc]Tc-TECANT-1: 5.74 ± 0.01 min Different operator [^99m^Tc]Tc-TECANT-1: 5.71 ± 0.01 min Different day [^99m^Tc]Tc-TECANT-1: 5.74 ± 0.02 min CV < 0.5%

* The lowest activity in the [^99m^Tc]Tc-TECANT-1 kit is expected to be 750 MBq in a volume of 1 mL in order to obtain the required activity of >500 MBq for a patient investigation. This sample was diluted 1:5 for radio-(RP)HPLC analysis according to Appendix A. Dilution series for LOQ determination and range was prepared from a solution of [^99m^Tc]TcO_4_^−^ to mimic the main impurity that needs to be quantified with the LOQ. ** equals 150 kBq/mL or 0.1% of a kit labelled with 750 MBq and diluted 1:5 for analysis. T = % [^99m^Tc]Tc-TECANT-1 (HPLC).

**Table 2 pharmaceutics-15-00885-t002:** Summary table of validation parameters, acceptance criteria and results for iTLC method for determination of the percentage of ^99m^Tc-colloidal species (B).

Parameter	Method	Acceptance Criteria	Result
Accuracy (B)	Comparison of R_f_ value of [^99m^Tc]Tc-TECANT-1	R_f_ sd < 0.1	mean = 0.82 sd = 0.02
Precision (B)	Analysis of ^99m^Tc-colloid species (n = 6)	RCP sd < 0.5%	mean: 2.97% sd: 0.32%
Specificity	R_f_-value [^99m^Tc]Tc-TECANT-1 R_f_-value ^99m^Tc-colloid species	>0.6 <0.4	0.80–0.85 (n = 6) 0.01–0.04 (n = 6)
LOQ (B)	Analysis of dilution of [^99m^Tc]TcO_4_^−^ eluate (0.8–800 MBq/mL)	>1%	LOQ = 1.6 MBq/mL LOD = 0.8 MBq/mL
Linearity (B)	Analysis of dilution of ^99m^Tc-solution (0.8–800 MBq/mL)	>0.9	0.9977
Range (B)	Analysis of dilution of ^99m^Tc-solution and calculate % of LOQ from maximum activity	<1%	0.2% (LOQ = 1.6 MBq/mL Max = 800 MBq/mL)
Robustness	R_f_ of ^99m^Tc-colloid species different day and operator	R_f_ < 0.4	0.05–0.12 (n = 6)

B = % of ^99m^Tc -colloid species (iTLC).

**Table 3 pharmaceutics-15-00885-t003:** Summary of initial ^99m^Tc-labelling experiments: influence of SnCl_2_ × 2H_2_O and trisodium citrate dihydrate amounts on radiochemical purity (radiolabelling: RT, 30 min, 400 MBq [^99m^Tc]TcO_4_^−^).

TECANT-2 Content [µg]	SnCl_2_ × 2H_2_O Content [µg]	Trisodium Citrate Dihydrate Content [mg]	% [^99m^Tc]Tc–TECANT-2 [HPLC]
10	5	0.13	93.6
10	10	0.13	93.8
10	15	0.13	93.2
10	20	0.13	88.0
10	15	0.26	92.5
10	15	0.39	88.8

**Table 4 pharmaceutics-15-00885-t004:** Kit composition of development batches and radiochemical purity of initial radiolabelling (radiolabelling: RT, 30 min, 600 MBq [^99m^Tc]TcO_4_^−^).

	Production Details						Quality Control Results		
	Vial 1			Vial 2		Vial 3	[^99m^Tc]Tc–TECANT-1 [%] HPLC	^99m^Tc-Colloid Species [%] iTLC	RCP [%]
Batch No.	TECANT-1 [µg]	SnCl_2_ × 2H_2_O [µg]	Trisodium Citrate dihydrate [mg]	Na_2_HPO_4_ × 12H_2_O [µg]	NaOH [mg]	NaH_2_PO_4_ × 2H_2_O [mg]			
05a/20	10 ^a^	20	0.13 ^b^	1.77	0.4	1.6	89.3 ± 1.1	n.d.	n.d.
05/20	10	15	0.13	1.77	0.4	1.6	95.3 ± 0.6	6.3 ± 0.3	89.0 ± 0.4
06/20	15	15	0.13	1.77	0.4	1.6	93.6 ± 0.0	4.7 ± 0.6	88.9 ± 0.5
07/20	50	15	0.13	1.77	0.4	1.6	93.0 ± 0.8	5.3 ± 1.4	87.7 ± 1.1
08/20	20	20	0.13	1.77	0.4	1.6	91.6 ± 0.1	10.8 ± 1.8	80.8 ± 1.6
09a/20	30	15	0.13	1.77	0.4	1.6	94.4 ± 0.2	3.5 ± 0.5	90.9 ± 0.3
09b/20	40	15	0.13	1.77	0.4	1.6	97.0 ± 0.3	7.9 ± 1.1	89.1 ± 1.0
10/20	20	15	0.13	1.77	0.4	1.6	99.0 ± 0.2	5.1 ± 0.3	93.9 ± 0.2

n.d.: not determined. ^[a]^ TECANT-2 was used in this batch. ^[b]^ Sodium citrate was part of vial 2 in this particular batch.

**Table 5 pharmaceutics-15-00885-t005:** Composition of the TECANT-1 kit for radiopharmaceutical preparation.

Name of ingredients	Amount	Function	Reference to Standards
Vial 1			
TECANT-1 TFA salt	20 μg (net peptide)	Active substance	/
Trisodium citrate dihydrate	0.13 mg	Pre-chelator	Ph. Eur., current valid edition
Stannous chloride dihydrate (SnCl_2_ × 2H_2_O)	15 μg	Excipient Reducing agent	Ph. Eur., current valid edition
Nitrogen	q.s.	Protective gas	Ph. Eur., current valid edition
Vial 2			
Disodium phosphate (Na_2_HPO_4_ × 12H_2_O)	4.5 mg	Excipient Buffer component	Ph. Eur., current valid edition
Sodium hydroxide (NaOH)	0.4 mg	Excipient Buffer component	Ph. Eur., current valid edition
Nitrogen	q.s.	Protective gas	Ph. Eur., current valid edition
Vial 3			
Sodium phosphate dihydrate (NaH_2_PO_4_ × 2H_2_O)	1.56 mg	Excipient Buffer component	Ph. Eur., current valid edition
Nitrogen	q.s.	Protective gas	Ph. Eur., current valid edition

q.s.: quantum satis.

**Table 6 pharmaceutics-15-00885-t006:** Batch analysis—test and requirements, methods and batch results for batches 01B/21, 02B/21 and TEC-01/01/22.

		BATCH No.		
Test	Requirements	01B/21	02B/21	TEC-01/01/22
**Lyophilisate**				
Appearance	White freeze-dried powder			
Vial 1		Conforms	Conforms	Conforms
Vial 2		Conforms	Conforms	Conforms
Vial 3		Conforms	Conforms	Conforms
Identity *Vial 1*	Retention time R_T_ of the sample complies with the standard R_T_ (R_T(TECANT-1)_ ÷ R_TS_ 1 ± 0.05)	1.04	1.01	1.00
Assay *Vial 1*	20 µg ± 2 µg	18.4 µg	18.7 µg	19.9 µg
Sn content (as SnCl_2_ × 2H_2_O)	15 µg ± 1.5 µg	15.1 µg	15.1 µg	15.4 µg
pH after reconstitution with 1 mL of water for injection				
Vial 1	6.4–7.4	6.7	6.4	6.6
Vial 2	10.7–11.7	10.4	11.2	10.8
Vial 3	4.5–5.5	5.1	4.9	5.0
Water content *Vial 2*	≤10%	7.0%	5.5%	9.2%
Sterility				
Vial 1	Sterile	Sterile	Sterile	Sterile
Vial 2	Sterile	Sterile	Sterile	Sterile
Vial 3	Sterile	Sterile	Sterile	Sterile
Bacterial endotoxins	<20 EU/kit			
Vial 1		0.5 EU/kit	0.75 EU/kit	0.75 EU/kit
Vial 2		0.5 EU/kit	0.75 EU/kit	0.75 EU/kit
Vial 3		0.5 EU/kit	0.75 EU/kit	0.75 EU/kit
**Kit after ^99m^Tc-labelling**				
% [^99m^Tc]Tc-TECANT-1 (HPLC) [%]	≥95	99.1 ± 0.5	99.7 ± 0.1	99.8 ± 0.2
%^99m^Tc-colloid species [%]	≤7	6.9 ± 1.2	3.2 ± 0.5	2.0 ± 0.5
RCP [%]	≥90	92.2 ± 0.6	96.5 ± 0.5	97.8 ± 0.5
pH after ^99m^Tc-labelling	6.0–8.0	7.2	7.5	7.3

**Table 7 pharmaceutics-15-00885-t007:** Summary of radiolabelling of batches 01B/21 and 02B/21 after different storage times.

		Results				
Test	Requirements	0	3 m	6 m	9 m	12 m
**BATCH No. 01B/21**						
% [^99m^Tc]Tc-TECANT-1 (HPLC) [%]	≥95	99.1 ± 0.5	99.6 ± 0.2	99.9 ± 0.1	99.5 ± 0.2	99.9 ± 0.1
% ^99m^Tc-colloid species [%]	≤7	6.9 ± 1.2	5.8 ± 0.4	6.8 ± 1.1	4.5 ± 0.6	6.5 ± 0.5
RCP [%]	≥90	92.2 ± 0.6	93.8 ± 0.3	93.2 ± 0.9	95.1 ± 0.4	93.4 ± 0.5
pH after ^99m^Tc-labelling	6.0–8.0	7.2	n.d.	n.d.	n.d.	7.2
**BATCH No. 02B/21**						
% [^99m^Tc]Tc-TECANT-1 (HPLC) [%]	≥95	99.7 ± 0.1	99.7 ± 0.03	99.7 ± 0.1	99.9 ± 0.1	99.9 ± 0.1
% ^99m^Tc-colloid species [%]	≤7	3.2 ± 0.5	1.7 ± 0.1	5.1 ± 0.3	5.4 ± 0.9	4.9 ± 0.6
RCP [%]	≥90	96.5 ± 0.5	98.0 ± 0.2	94.5 ± 0.3	94.5 ± 0.5	95.2 ± 0.6
pH after ^99m^Tc-labelling	6.0–8.0	7.5	n.d.	n.d.	n.d.	7.3

n.d.: not determined.

**Table 8 pharmaceutics-15-00885-t008:** Summary of the stability study of the radiolabelled compound prepared from batches 01B/21 and 02B/21.

	Results (mean ± SD)			
Parameters	0 h	1 h	2 h	4 h
**BATCH No. 01B/21** (Activity range: ~1000–1400 MBq, labelling volume: 0.5–1.5 mL, final molar radioactivity range: 67–95 GBq/µmol), n = 6				
Appearance	Conforms	Conforms	Conforms	Conforms
pH 7–8	Conforms	n.d.	n.d.	n.d.
% [^99m^Tc]Tc-TECANT-1 (HPLC) [%]	98.9 ± 0.7	99.4 ± 0.6	98.5 ± 0.3	96.3 ± 1.0
% ^99m^Tc-colloid species [%]	2.2 ± 1.1	4.2 ± 1.1	4.6 ± 1.1	5.1 ± 1.7
RCP [%]	96.7 ± 0.7	95.2 ± 1.2	94.0 ± 1.3	91.4 ± 2.3
**BATCH No. 02B/21** (Activity range: ~900–1100 MBq, labelling volume: 0.5–0.9 mL, final molar radioactivity range: 60–74 GBq/µmol), n = 3				
Appearance	Conforms	Conforms	Conforms	Conforms
pH 7–8	Conforms	n.d.	n.d.	n.d.
% [^99m^Tc]Tc-TECANT-1 (HPLC) [%]	98.6 ± 0.5	98.3 ± 1.0	96.7 ± 0.3	95.5 ± 0.4
% ^99m^Tc-colloid species [%]	2.9 ± 1.0	3.6 ± 1.7	3.7 ± 1.4	4.0 ± 1.1
RCP [%]	95.8 ± 0.54	94.8 ± 0.7	93.8 ± 0.3	92.3 ± 0.2

## Data Availability

The data presented in this study are available in this article and Appendix A.

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
