# Peer review of "Development of the 99mTc-Labelled SST2 Antagonist TECANT-1 for a First-in-Man Multicentre Clinical Study"

_pharmaceutics, 2023, doi:10.3390/pharmaceutics15030885_

Round 1
Reviewer 1 Report
The aim of the study is to provide a reliable kit for preparation of [99mTc]Tc-TECANT-1, as a novel high potential agent for SPECT imaging of SST2 overexpressing tumors in a hospital radiopharmacy setting. The study is well designed and conducted and the manuscript is well written. The manuscript could be attractive to the readers and therefore it is recommended for publication in the Journal. However, the following suggestions are also sent to increase the quality of the manuscript:
ü Please specify the specific activity to [99mTc]Tc-TECANT-1 in the final formulation
ü Please tell more about the properties of the [99mTc]TcO4- eluate (including pH, …) used for radiolabeling.
ü In 2-vial kit method, may the volume of the reaction effect on the radiochemical purity?
Author Response
- Please specify the specific activity to [99mTc]Tc-TECANT-1 in the final formulation
We have added the molar radioactivity range in Table 8 for 2 batches in the final formulation (rather than specific activity, thereby following the nomenclature guidelines as required by the journal)
- Please tell more about the properties of the [99mTc]TcO4- eluate (including pH, …) used for radiolabeling.
We now have added a more detailed description of the eluate used and the 2 types of generators
- In 2-vial kit method, may the volume of the reaction effect on the radiochemical purity?
Indeed, this could have an influence on the RCP, however, as the results were not satisfiying, but could be resolved with the 3 vial kit, we did not do any more experiments to verify this. So we have no data to substantiate that and therefore any discussion would only be speculative, so we have avoided this.
Reviewer 2 Report
The comments are in the word file attached herewith this message.

Author Response
Major comments: The present article depicts in details, the manufacturing of a freeze-dried kit (a set of three vials) for preparation of 99mTc-labeled-TECANT-1, a SSTR2 antagonist. Though the title of the manuscript indicates the development of the kit for a first in-man multicentre-clinical study, no clinical data is presented in this article. The article lacks the novelty; however, an extensive parameter variation has been performed for optimization of the freeze-dried kit for radio-labeling with 99mTc and the same is the strength of this article. I request authors to kindly revise the manuscript as per the comments given below before it could be considered for publication.
- Page 3, Figure 1, the authors have drawn the 99mTc-complex with N4 group. Authors are request to kindly take care of deprotonation of nitrogen atoms in the complex structure and can re-draw it accordingly.
Sorry, but this is not correct. The structure of the Tetramine complex with Technetium is known and has been well characterised and is (interestingly) not a deprotonated form but the octahedral trans-dioxo Tc(V) monocationic structure without deprotonation, this has been described in several papers and we had also confirmed this with the TECANT peptides.
Following reference can be consulted:
Tetraamine-coupled peptides and resulting (99m)Tc-radioligands: an effective route for receptor-targeted diagnostic imaging of human tumors. Nock B, Maina T. Curr Top Med Chem. 2012;12(23):2655-67. doi: 10.2174/1568026611212230003. In this paper references 53-63 refer to this structural confirmation in initial original publications:
e.g.: doi:10.1021/ic50222a006; https://dx.doi.org/10.1016/S0020-1693(00)85099-9; https://doi.org/10.1016/0277-5387(96)00240-9; DOI: 10.1080/00958970108022641 and many more….
We have, however slightly changed the figure to clarify the cationic structure of the complex (as was made in other, original papers)
- The authors have demonstrated the formulation of freeze-dried kits for both TECANT-1 and TECANT-2, however toxicity studies were performed for TECANT-1 only. To avoid the confusion to the readers, authors are kindly requested to mention in the beginning of the concerned sub-heading (in Experimental section) about what they are going to perform in this section.
We have added TECANT 1 to the respective sub-heading for clarification
- Page 9, sub-heading 3.2. the optimization studies involving the variation in the amount of the excipients has been performed with TECANT-2 whereas kit radio-labeling conditions have been optimized with TECANT-2. Why authors have decided to perform half of the optimization studies with TECANT-1 and the other half with TECANT-2? Kindly clarify.
Indeed, we understand that this may cause confusion. The reason is that we did initial experiments with TECANT 2, simply for economical reasons. TECANT 1 was available in GMP quality (which we wanted to use for reasons of traceability to the final clinical product with the same batch. To have sufficient amounts for these main steps, we used for initial evaluations TECANT 2, which from a scientific perspective, has the overall same labelling properties as TECANT 1. We have tried to clarify this in the initial description by adding a separate point 2.3. on “study design”.
- Also, since the optimized freeze-dried kit formulation for 99mTc radiolabeling is meant for TECANT-1 only, by bringing in the TECANT-2 for few studies authors have created some confusion in the manuscript. I advise authors to mention the text related to their planning of experimental section at the beginning itself (either in the last paragraph of introduction or in the Experimental section). This will help to clarify the objective of the experimental work to the readers.
We agree and have now added this at the beginning in the experimental section as a separate point (see above)
Minor comments:
- Authors have designated Figure number 1 to the first figure in the materials and methods section. But again, in the Results section, a figure has appeared bearing the same figure number. Kindly correct.
We have corrected this
|
Page number |
Line number |
Comment |
|
3 |
97 |
‘due to’ could be changed to ‘corresponding to’ |
|
3 |
114 |
Kindly put the minus (–) sign in superscript position (in rest of the manuscript also) |
|
7 (Results section) |
Figure 1 |
Kindly re-number the Figure 1 as Figure 2 Also, the Y-axis of the figure is representing values in mV (generally given for UV profile) however caption depicts it as radio-HPLC profile, in latter case the Y-axis should be represented in CPS (counts per second or equivalent units) |
We have corrected all as indicated above
Reviewer 3 Report
This paper is devoted to the development of a method for facile preparation of a 99mTc-labelled SSTR2 antagonist, [99mTc]Tc-TECANT-1, in a hospital radiopharmacy setting. Despite the successful use of the PET in SSTR diagnostics and, in particular, [68Ga]Ga-DOTA-TOC (SomaKit TOC®) and [68Ga]Ga-DOTA-TATE (NETSPOT™), the isotope 99mTc remains the most demand in nuclear medicine. Therefore, the relevance of the work does not raise questions. The design of the experiment is carefully planned. The results obtained correspond to the stated goals. At the same time, there are some comments to work on:
1) The literature review should be expanded by the information on kits for the SSTR diagnosis by PET and SPECT. In particular, the SomaKit TOC® and NETSPOT™ kits for the 68Ge/68Ga generator will compete with this kit.
2) In Section 3.1, in addition to radio-HPLC chromatograms, radio-TLC data should also be added because two analytical procedures are validated to determine radiochemical purity.
3) Section 2.4 describes the preparation of the kit. For preparation of vial 1, 1.5 mg SnCl2×2H2O add in 1.5 ml ethanol. However, Table 6 lacks a test for ethanol residual amount and acceptability of its criteria. The authors should justify the absence of these criteria or add data on the analysis of ethanol in the dissolved lyophilizate by gas chromatography.
4) According to the data obtained, the optimal pH value for the reaction is 11. However, the authors used two generators manufactured by Polatom and Curium with different pH values according to the specification (5.5–7.5 and 4.0–8.0, respectively). In the text (explanations to the table), authors should indicate which generator was used for each case. In addition, have any experiments been carried out on the effect of the generator on the efficiency of complex formation?
5) The optimal value of the activity of sodium pertechnetate is 600 MBq since RCP decreased with increasing activity. Have radiochemical experiments been performed in the presence of stabilizers (e.g. sodium ascorbate or sodium gentisinate)?
6) Have any experiments been carried out with the activity of sodium pertechnetate 1200 MBq and a twofold increase in all components of the kit?
Author Response
This paper is devoted to the development of a method for facile preparation of a 99mTc-labelled SSTR2 antagonist, [99mTc]Tc-TECANT-1, in a hospital radiopharmacy setting. Despite the successful use of the PET in SSTR diagnostics and, in particular, [68Ga]Ga-DOTA-TOC (SomaKit TOC®) and [68Ga]Ga-DOTA-TATE (NETSPOT™), the isotope 99mTc remains the most demand in nuclear medicine. Therefore, the relevance of the work does not raise questions. The design of the experiment is carefully planned. The results obtained correspond to the stated goals. At the same time, there are some comments to work on:
1) The literature review should be expanded by the information on kits for the SSTR diagnosis by PET and SPECT. In particular, the SomaKit TOC® and NETSPOT™ kits for the 68Ge/68Ga generator will compete with this kit.
We have added reference to new Ga-68 kits as potential competitors with the described product
2) In Section 3.1, in addition to radio-HPLC chromatograms, radio-TLC data should also be added because two analytical procedures are validated to determine radiochemical purity.
We have added an example of TLC in the supplementary materials as Figure S1
3) Section 2.4 describes the preparation of the kit. For preparation of vial 1, 1.5 mg SnCl2×2H2O add in 1.5 ml ethanol. However, Table 6 lacks a test for ethanol residual amount and acceptability of its criteria. The authors should justify the absence of these criteria or add data on the analysis of ethanol in the dissolved lyophilizate by gas chromatography.
The amount of Ethanol concentration was 1.5% in the solution dispensed in kit vials, these were undergoing freeze drying in 1mL fractions, containing 15µl of Ethanol which is irrelevant in terms of total Ethanol amount. In the freeze drying process Ethanoil is removed withg the water and determination of the water content ensures removal of Ethanol. This risk based approach has also been accepted by three different pharmaceutical authorities. We have added a short statement on this approach for clarification.
4) According to the data obtained, the optimal pH value for the reaction is 11. However, the authors used two generators manufactured by Polatom and Curium with different pH values according to the specification (5.5–7.5 and 4.0–8.0, respectively). In the text (explanations to the table), authors should indicate which generator was used for each case. In addition, have any experiments been carried out on the effect of the generator on the efficiency of complex formation?
Indeed, this is a valuable point. We now have added data of using the same kit-batch with 2 different generators , radiolabelled under the same conditions without any significant difference in the results. This has been added as a table in supplementary information Tab S8
5) The optimal value of the activity of sodium pertechnetate is 600 MBq since RCP decreased with increasing activity. Have radiochemical experiments been performed in the presence of stabilizers (e.g. sodium ascorbate or sodium gentisinate)?
We have not included stabilizers at this stage, the formulation intended for a Phase I study, of course we agree that for further development for wider use, this is an important factor to be evaluated. We have added a sentence on this in the manuscript
6) Have any experiments been carried out with the activity of sodium pertechnetate 1200 MBq and a twofold increase in all components of the kit?
We have not included stabilizers at this stage, the formulation intended for a Phase I study, of course we agree that for further development for wider use, this has to be still evaluated. We have added a sentence on this in the manuscript